# Lignin-Modifying Enzymes in *Scedosporium* Species

**DOI:** 10.3390/jof9010105

**Published:** 2023-01-12

**Authors:** Wilfried Poirier, Jean-Philippe Bouchara, Sandrine Giraud

**Affiliations:** University of Angers, University of Brest, IRF (Infections Respiratoires Fongiques), SFR ICAT 4208, CEDEX 9, 49933 Angers, France

**Keywords:** *Scedosporium*, lignin, peroxidases, oxidases, modifying enzymes

## Abstract

*Scedosporium* species are usually soil saprophytes but some members of the genus such as *S. apiospermum* and *S. aurantiacum* have been regularly reported as causing human respiratory infections, particularly in patients with cystic fibrosis (CF). Because of their low sensitivity to almost all available antifungal drugs, a better understanding of the pathogenic mechanisms of these fungi is mandatory. Likewise, identification of the origin of the contamination of patients with CF may be helpful to propose prophylactic measures. In this aim, environmental studies were conducted demonstrating that *Scedosporium* species are abundant in human-made environments and associated with nutrient-rich substrates. Although their natural habitat remains unknown, there is accumulated evidence to consider them as wood-decaying fungi. This study aimed to demonstrate the ability of these fungi to utilize lignocellulose compounds, especially lignin, as a carbon source. First, the lignolytic properties of *Scedosporium* species were confirmed by cultural methods, and biochemical assays suggested the involvement of peroxidases and oxidases as lignin-modifying enzymes. *Scedosporium* genomes were then screened using tBLASTn searches. Fifteen candidate genes were identified, including four peroxidase and seven oxidase genes, and some of them were shown, by real-time PCR experiments, to be overexpressed in lignin-containing medium, thus confirming their involvement in lignin degradation.

## 1. Introduction

Fungi are key players in the degradation of lignocellulosic biomass. Besides cellulose and hemicelluloses (mainly xylans), lignin is a macromolecule providing strength as well as rigidity to the plant cell wall and protecting its two polysaccharidic partners against microbial attack by hydrolytic enzymes. Meanwhile, the degradation of cellulose and hemicellulose involves glycosidases (mainly cellobiohydrolases, cellulose cellobiosidases and glycosidases, or xylanases and xylohydrolases, respectively), degradation of lignin is more complex as it is a polyphenolic macromolecule resulting from the polymerization of three types of monolignols (paracoumarylic, coniferylic and sinapylic acids). As a consequence, whereas many microorganisms are able to use cellulose and hemicellulose as the carbon source, only a few groups of fungi are able to degrade lignin, among which the wood-decaying Basidiomycetes, also termed “white-rot fungi”, are the most efficient. They mineralize the complex lignin polymer by the synergistic action of several extracellular enzymes, such as peroxidases and laccases [1].

During the past few decades, the metabolic pathways and the enzymatic mechanisms involved in lignin degradation by Basidiomycetes have been extensively studied [2,3]. Conversely, little is known about the lignocellulose degradation by Ascomycetes. Wood decay by some Ascomycetes was first observed in 1954 by Savory [4,5]. These fungi were designated as “soft-rot fungi” due to their ability to degrade cellulose in notable amounts compared to lignin. Nevertheless, several studies demonstrated that some Ascomycetes may also attack lignin. They are able to mineralize lignin or lignin model compounds, but to a much lower extent than the wood-inhabiting white-rot fungi [6,7,8,9,10,11]. Laccases seem to be key elements in the degradation process. Indeed, in most of the studies, these fungi have been reported to produce laccases whereas no peroxidase activity was detected. Moreover, whereas lignin degradation is considered as secondary metabolism in white-rot fungi, the Sordariomycete *Phialemonium inflatum* (formerly *Paecilomyces inflatus*) mineralizes lignin during its initial growth stage, thus during primary metabolism [12].

The *Scedosporium* species (formerly known as the asexual state of *Pseudallescheria* species) are soil saprophytic filamentous fungi that are known to determine severe disseminated infections in immunocompromised patients by inhalation of some airborne spores, but also to colonize the respiratory tract of immunocompetent individuals with chronic pulmonary disease. They are notably the second filamentous fungi in terms of frequency in the respiratory tract of patients with cystic fibrosis. These Sordariomycetes exhibit a low susceptibility to current antifungal drugs, and a better understanding of their pathogenic mechanisms is required to identify new therapeutic targets. Likewise, a better knowledge of the origin of the contamination of patients should allow us to propose prophylactic measures. In this context, several environmental studies have been performed which demonstrated that *Scedosporium* species are mostly found in polluted environments and anthropogenic areas such as agricultural soils, gardens, wastewaters, playgrounds or city parks [13,14,15]. Nevertheless, the natural habitat of these fungi remains to be determined. They are capable to grow in environments with a high osmotic pressure or in poorly aerated soils, to tolerate high salt concentrations [16], and to degrade hydrocarbons. In addition, there is now accumulated evidence to consider them as wood-decaying fungi.

Since it has been reported that filamentous fungi may use the lignin modifying pathway to degrade complex molecules [17,18,19], this work aimed to analyze the lignocellulolytic properties of *Scedosporium* species.

## 2. Materials and Methods

### 2.1. Isolates and Culture Conditions

The study was conducted on three *Scedosporium* species. Three isolates per species (including one whole-genome sequenced) were used: *S. apiospermum* IHEM 14462, IHEM 23580 and UA 110350824; *S. aurantiacum* IHEM 23578, UA 100353192-01 and UA 110344103; and *S. dehoogii* UA 120008799, UA 110354504 and UA 110354521. Isolates were maintained by regular passages on YPDA plates (containing in g per liter: yeast extract, 5; peptone, 10; dextrose, 20; agar, 20; and chloramphenicol, 0.5).

Growth studies were carried out in triplicate on a synthetic agar-based medium derived from the Scedo-Select III selective culture medium [20] and containing in g per liter: carbon source, 0.9; ammonium sulphate, 5; potassium dihydrogenophosphate, 1.25; magnesium sulphate, 0.625; agar, 20; and chloramphenicol, 0.5. Different compounds were compared in this culture medium as the unique carbon source: glucose for control conditions, xylan (SERVA Electrophoresis GmbH, Heidelberg, Germany), cellulose (Sigma-Aldrich, Saint Louis, MI, USA) or lignin (Sigma-Aldrich). Mycelium from a 14-day-old YPDA culture was collected using a sterile needle and inoculated by central pricking. Plates were incubated at 37 °C and growth was evaluated by measuring the diameter of the colonies every day for ten days. Results were compared with those obtained on the rich and commonly used medium, YPDA. For statistical analyses, an ANOVA–Tukey test was performed to compare growth at 10 days within each culture conditions, * *p*-value < 0.05.

For other experiments, isolates were grown on Potato Dextrose Agar (Conda, Madrid, Spain) plates at 37 °C for seven days to enhance sporulation. Conidia were harvested from colonies by aseptically scraping the plates in water and filtration through Miracloth^®^ mesh filter (Merck, Darmstad, Germany) to remove the hyphae, and finally enumerated by hemocytometer counts. To evaluate the impact of the carbon source on the expression level of the target genes, a short duration of incubation (the same for all carbon sources) was needed to avoid degradation of mRMAs. As the germination kinetic vary according to the carbon source, a normalization of the germination step therefore was performed by preincubation of the conidia in YPD medium for 24 h in order to obtain germ tubes. Briefly, 2 × 10^7^ conidia were inoculated in 50 mL of YEPD medium (containing in g per liter: yeast extract, 5; peptone, 10; dextrose, 20; and chloramphenicol, 0.5). After a 24 h-incubation at 37 °C with agitation (120 rpm), germ tubes collected on 11-μm pore-size nylon filters were inoculated in 50 mL of derived Scedo-Select III broth (containing the same components as the agar-based culture medium, except agar) with the appropriate carbon source, which was then incubated with agitation (120 rpm) at 37 °C during 4 h for RNA extraction or 24 h for enzymatic assays. 

### 2.2. Enzymatic Assays

Culture supernatants were collected after 24 h of incubation and preserved by freeze-drying before lyophilization. Lyophilizates were resuspended in 1 mL of saline and used for two-point enzymatic assays. For each enzymatic assay, three independent experiments were realized and all reactions were carried out in duplicate. Protein concentration was determined by fluorimetry using Qubit. The specific activity corresponds to the number of international enzyme units (i.e., µmoles of substrates converted per unit time) per mg of total proteins. According to the data distribution, statistical analysis was performed using a Student’s *t*-test or Mann–Whitney test.

#### 2.2.1. Detection of Peroxidase Activity

Detection of peroxidase activity was performed using the *O*-PhenyleneDiamine (OPD; Sigma-Aldrich) oxidation test [21]. Two different controls were used: (i) a control lyophilizate obtained from a fungus-free culture medium submitted to the same incubation time and temperature; (ii) a control lyophilizate supplemented with 2.5 μL of horseradish peroxidase-conjugated antibodies (0.5 ng/µL).

The reaction mixture contained OPD (0.75 mg/mL final concentration), 5 μL of hydrogen peroxide 30%, 12.5 μL of lyophilizate and an appropriate volume of 0.1 M citric acid/sodium phosphate buffer pH 5 to reach a final volume of 250 µL. The mix was incubated at room temperature for 5 min, and the reaction was stopped with 62.5 µL of 0.2 M sulfuric acid. OPD oxidation yielded a soluble end-product which was yellow-orange in color. The absorbance was determined at 490 nm. The extinction coefficient used was 1.578 mM^−1^cm^−1^, as described previously [22].

#### 2.2.2. Detection of Oxidase Activity

The ability of the isolates to produce extracellular oxidases was quantified using 2-2′-Azinobis(3-ethylBenzoThiazoline-6-Sulphonic acid) (ABTS; Sigma-Aldrich) as described by Shrestha et al. [23]. ABTS working solution consisted in 0.1 mM ABTS and hydrogen peroxide 0.0075% in 0.1 M citric acid/sodium phosphate buffer pH 5. The reaction solution was composed by equal volumes (i.e., 150 μL) of ABTS working solution and samples. A negative control (obtained from a fungus-free culture medium) and a positive control (the same supplemented with a suitable amount of *Aspergillus* laccase from Sigma-Aldrich) were included in each series of tests. All reactions were incubated at 37 °C for 20 min. The oxidation of ABTS was followed by measurement of the absorption at 420 nm. The molar extinction coefficient used was 36 mM^−1^cm^−1^.

### 2.3. Genome Mining

A literature review was performed to identify lignocellulolytic fungi and then focused on lignin-modifying enzymes. Orthologs of the genes identified by this way were searched in *Scedosporium* genomes (JOWA01000000 for *S. apiospermum*, JUDQ01 for *S. aurantiacum* and PGIR00000000.1 for *S. dehoogii*) through tBLASTn analysis (https://blast.ncbi.nlm.nih.gov/Blast.cgi, accessed on 10 January 2023). Only results with an E-value < 1 × 10^−6^ on at least 40% of the query sequence were considered. Gene organization (exon/intron) was determined using HMMER program on EnsemblFungi (https://fungi.ensembl.org/hmmer/index.html, accessed on 10 January 2023) and validated with Augustus (https://bioinf.uni-greifswald.de/augustus/, accessed on 10 January 2023).

For classification analysis, all peroxidase sequences were collected from PeroxiBase (http://peroxibase.toulouse.inra.fr, accessed on 10 January 2023) [24], and multicopper oxidase (MCO) sequences from the NCBI GenBank [25]. The Geneious software package [26] was used for the phylogenetic study. An alignment was created using the default settings for multiple alignments. Phylogenetic trees were constructed by the neighbor-joining method. Bootstrapping was carried out with 500 replications.

### 2.4. RNA Isolation, Reverse Transcription and Real-Time Quantitative PCR

Fungal cells were ground in liquid nitrogen with a mortar and pestle. Total RNAs were extracted using the NucleoSpin^®^ RNA plant kit (Macherey-Nagel, Dueren, Germany), according to the manufacturer’s instructions. RNA samples were then treated with 2 U of RNase-free DNase I (Ambion^TM^ Life Technologies, Carlsbad, CA, USA), following the manufacturer’s recommendations. RNA quantity and quality were evaluated by Qubit dosage and agarose-gel electrophoresis, respectively. Complementary DNA (cDNA) was synthetized from 500 ng total RNA using High Capacity cDNA Reverse Transcription kit (Applied Biosystems, Foster City, CA, USA), and random primers, according to the protocol supplied by the manufacturer. Thereafter, cDNA samples were 10-fold diluted and used as template for real-time quantitative PCR (qPCR). PCR reactions were performed in a final volume of 12.5 μL containing FAST SyBR^®^Green PCR Master Mix (Applied Biosystems), 200 nM of each primer (Integrated DNA Technologies Inc., Leuven, Belgium) and 2 μL of diluted cDNA. Primers used for qPCR experiments and PCR efficiencies are compiled in Appendix A, respectively. qPCR reactions were carried out on a StepOnePlus^TM^ thermocycler (Applied Biosystems) with the following amplification program: 95 °C for 20 s, 40 cycles of 95 °C for 3 s, 60 °C for 30 s. Melting curve analysis (95 °C for 15 s and stepwise annealing from 60 to 95 with 0.3 °C increments) was performed immediately after the amplification. For each gene, fold changes relative to standard condition (i.e., in the presence of glucose as the carbon source) were calculated with the delta–delta Ct method [27,28]. Two reference genes were selected among nine candidates on the basis of their stable expression (validated by an ANOVA–Tukey statistical test) whatever the culture conditions and the species considered [29,30,31,32]. For each point, three biological replicates and two technical replicates were performed and a variation in expression of a given gene was considered significant if the log2 fold change ± standard deviation was <−1 or >1.

## 3. Results

Figure 1a illustrates the results obtained for *S. apiospermum* IHEM 14462 isolate. The fungus was not able to develop in the absence of any carbon source (data not shown), and it grew more slowly on a lignin-containing synthetic medium compared to the other culture media (Figure 1a and Appendix A). In addition, for most of the strains, the mean diameter of the colonies after ten days of incubation was significantly higher in the presence of cellulose or xylan compared to the results obtained in the presence of lignin or, strikingly, of glucose (Figure 1b). This suggests that these *Scedosporium* species are able to assimilate lignocellulosic compounds, including lignin. Therefore, we decided to further investigate this property. During the last few decades, the ability to attack lignin has been found to depend on two main types of enzymatic activities: peroxidase and laccase. In order to quantify these activities, enzymatic assays were performed using OPD and ABTS as substrates for peroxidase and laccase activities, respectively.

There was an important inter-experimental variability (possibly explained by the fact the analysis was conducted during the early phase of hyphal growth—after 24 h of incubation) leading to important standard deviations. Significant differences were observed for three of the strains tested regarding the peroxidase activity and three other strains for laccase activities (Figure 2). Nevertheless, for all the species both enzymatic activities tend to increase in the presence of lignin as carbon source.

Genes reported in the literature as encoding such enzymes were used to screen the genome of *S. apiospermum* strain IHEM 14462 by tBLASTn searches. Fifteen putative genes were identified, four encoding members of the peroxidase family and the others encoding genes belonging to the MCO superfamily. Among these genes, three were annotated as encoding uncharacterized or hypothetical proteins (SAPIO_CDS2597, SAPIO_CDS8589 and SAPIO_CDS9845) and three others were considered by Augustus as pseudogenes (SAPIO_CDS2438, SAPIO_CDS4198 and SAPIO_CDS4646) because of the lack of introns. To assess whether the genes predicted in silico were actually involved in lignin metabolism, we studied their expression in *Scedosporium* cells grown for 4 h in a lignin-containing synthetic liquid medium and in control conditions. In this aim, gene structure, especially the position of intronic sequences, was verified when possible and forward/reverse primers were designed from distinct exons (Appendix A). PCR efficiency was evaluated and considered as acceptable within a range of 80% to 120% (Appendix A).

As the 2^−∆∆CT^ method was used to analyze the relative gene expression, the first step was to select reference genes. Among the nine genes considered as the most interesting for qPCR studies in filamentous fungi [32], seven showed significant differences between the three *Scedosporium* species tested and two did not show significant variations whatever the culture conditions (Figure 3). Thus, the expression of actin and *β*-tubulin genes was used for normalization.

Similar expression patterns were obtained for *S. apiospermum* and *S. dehoogii* (Figure 4). All peroxidase genes, except SAPIO_CDS10447, were overexpressed compared to control conditions suggesting direct or indirect involvement in lignin degradation. In *S. aurantiacum*, only two of these genes (SAPIO_CDS4198 and SAPIO_CDS10583) were overexpressed. Considering the MCO superfamily, only SAPIO_CDS8659 was overexpressed in all species. Three other genes were also overexpressed but only in two species *S. apiospermum* and *S. dehoogii*: SAPIO_CDS2438, SAPIO_CDS0314 and SAPIO_CDS4646. Finally, SAPIO_CDS10367 was overexpressed only in *S. dehoogii*.

## 4. Discussion

This study reports for the first time the ability of *Scedosporium* species to use each component of lignocellulose (i.e., lignin, cellulose and xylan) independently as a carbon source. Even if results should be confirmed on natural lignin (kraft lignin was used in our experiments), they suggest that the *Scedosporium* species participates in biomass degradation, an essential role in the Earth’s carbon cycle. Two of the three ecological groups of lignocellulolytic fungi, the white- and brown-rot fungi, mainly belong to Basidiomycetes whereas wood-decaying Ascomycetes are usually included in soft-rot fungi. Basidiomycetes seem to be the only organism capable of an efficient lignin mineralization using specific class II peroxidases (lignin, manganese or versatile peroxidases). In contrast, little is known about the capability of Ascomycetes to attack the complex structure of lignin polymers. In a previous work, we demonstrate that the *Scedosporium* species displays all the genetic equipment needed for the intracellular degradation of phenolics’ compounds. These lower funneling pathways degrade aromatic compounds, such as those resulting from the extracellular degradation of lignin [33].

Results from our study are a first step for classifying *Scedosporium* species among the soft-rot fungi. Further characterization of the peroxidase arsenal analyzed in this study should allow definitive exclusion of these Ascomycetes from white-rot fungi. All *Scedosporium* peroxidases identified by genome mining belonged to class I peroxidase (Figure 5): two, encoded by SAPIO_CDS4198 and SAPIO_CDS10583, were catalase peroxidases, whereas the others belonged to the cytochrome c peroxidase (SAPIO_CDS3675) or the hybrid ascorbate-cytochrome c peroxidase (SAPIO_CDS10447) subclasses. The presence of a C-terminal WSC (cell wall integrity and stress response component) domain allows us to classify the latter among the hybrid-type B peroxidases [34].

Peroxidase and oxidase activities were detected for the three *Scedosporium* species studied, and they were shown to be markedly increased when *Scedosporium* isolates were cultivated in the presence of lignin, suggesting their involvement in lignin degradation. Previous studies suggested that Ascomycetes mainly used oxidases as lignolytic enzymes. Five out of the eleven putative genes of the MCO superfamily were overexpressed in at least one *Scedosporium* species, including two encoding unclassified related-MCO proteins, SAPIO_CDS4646 and SAPIO_CDS2438. The increased gene expression of the canonical MCO of FET3 subfamily SAPIO_CDS0314, the Abr-1-like protein encoded by SAPIO_CDS8659 and the “ferroxidase/laccase” encoded by SAPIO_CDS10367 suggests a link between the lignolytic properties and iron metabolism (Figure 6).

In brown-rot fungi, lignin chemical changes involving small oxidants are largely documented, the Fenton system being the best-known candidate for production of oxidant molecules. Fe^2+^ reacts with H_2_O_2_ to produce water, Fe^3+^ and usually a hydroxyl radical [35]. The use of extracellular reactive oxygen species (ROS) for lignolytic activity of fungi has been described since the middle of the last century. The mechanism has not been specified, but previous studies evidenced the involvement of ROS production in the attack of lignin in soft-rot fungi, and mineralization of synthetic lignin was decreased by the specific inhibition of hydroxyl radical production [35,36]. Proteins overexpressed in *Scedosporium* species in lignin-containing medium suggest that such mechanisms may be implicated. Nevertheless, complementary experiments (functional genomic studies and structural analysis of lignin by-products) are required to understand the role and functions of these proteins as well as the chemical reactions involved.

Although the natural habitat of *Scedosporium* species remains to be determined, indirect evidence suggest an association of the fungi with wood. Several strains of the genus *Scedosporium* were recovered from submerged woods in estuarian or marine coasts [37,38], from forest soils or wood [16,39] but also from xylophagous insects [40,41,42]. Recent works on Ascomycetes showed that these fungi use the same enzymatic arsenal to degrade lignocellulose and organic pollutants such as aromatic hydrocarbons [17,18,19]. The lignolytic ability of *Scedosporium* species may also explain their occurrence in polluted environments. In addition, a link between catabolism of hydrocarbons and fungal virulence, suggested for human fungal pathogens [43], was demonstrated in plant pathogenic fungi. For instance, Martins et al. [44] reported an increase in the pathogenic potential in human of some fungi upon exposure to aromatic chlorinated compounds. In plants, *Sclerotinia sclerotiorum* metabolizes salicylate, a key antifungal defense component of the host [45] and the virulence of *Botrytis cinerea* on grape is correlated at least in part with its ability to metabolize stilbene-type phytoalexins [46]. A better understanding of *Scedosporium* species adaptation to their environment and especially polluted environments may allow us to identify interesting pathways as targets for the development of more potent antifungal drugs.

## Figures and Tables

**Figure 1 jof-09-00105-f001:**
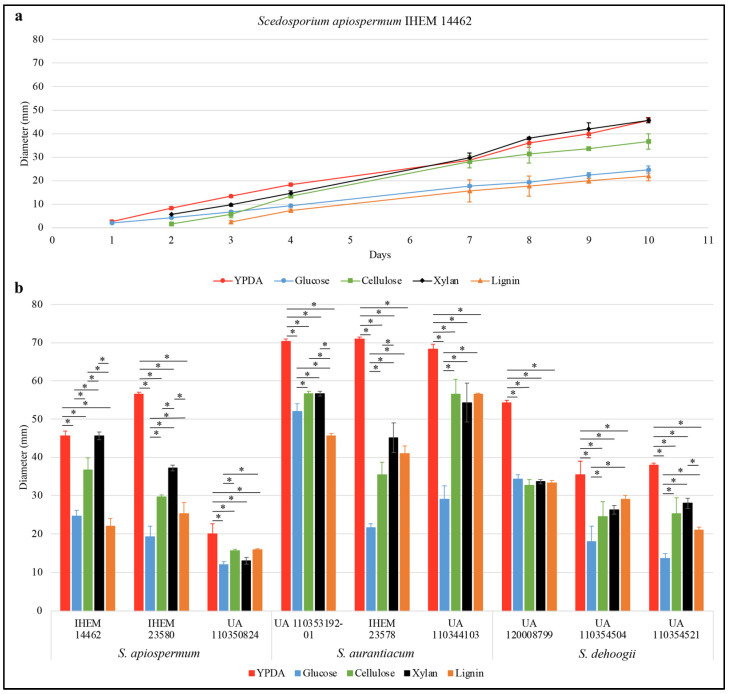
(**a**) Kinetics of growth of *Scedosporium apiospermum* IHEM 14462 on a synthetic agar-based medium Scedo-Select III containing a unique carbon source: glucose for control conditions, lignin, cellulose or xylan. (**b**) Colony diameter (mm) after 10 days of culture on agar-based media according to the carbon source. Fungal culture collections: IHEM, clinical strains; and UA, clinical or environmental strains. For statistical analyses, an ANOVA–Tukey test was performed to compare growth at 10 days within each culture conditions, * *p*-value < 0.05.

**Figure 2 jof-09-00105-f002:**
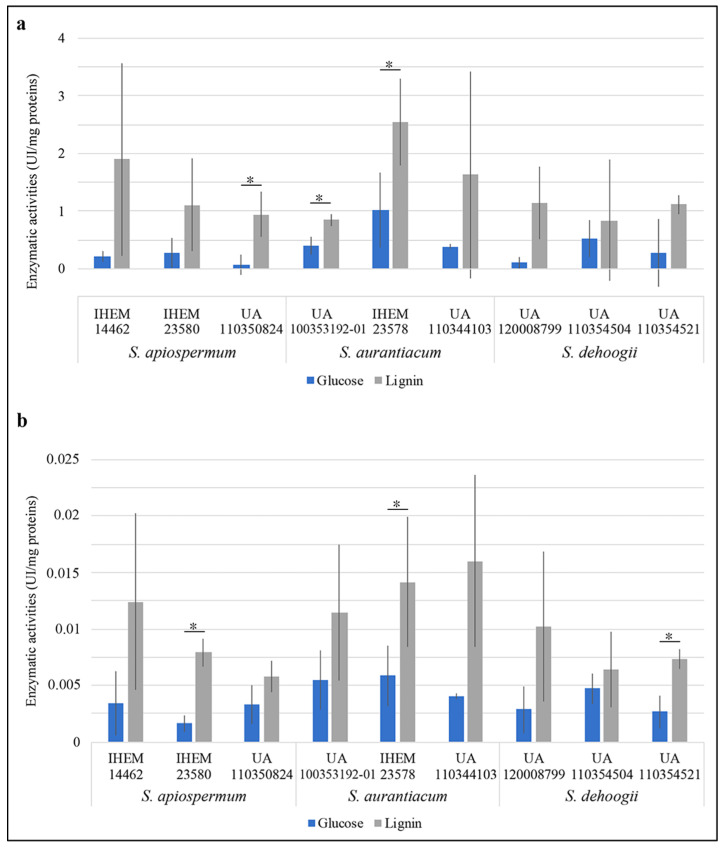
(**a**) Peroxidase and (**b**) oxidase activities (UI/mg proteins) of culture supernatants collected after 24 h of incubation. For each enzymatic assay, three independent experiments were realized and all reactions were carried out in duplicate. According to the data distribution, statistical analysis was performed using a Student’s *t*-test or Mann–Whitney test, * *p*-value < 0.05.

**Figure 3 jof-09-00105-f003:**
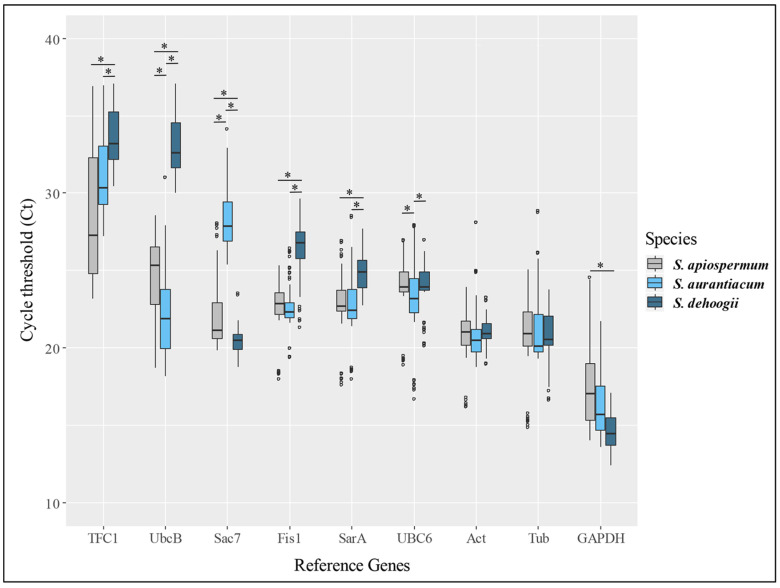
Variations in the expression of several reference genes in the different culture conditions studied. For each gene, an ANOVA–Tukey test was performed to analyze the variations between *Scedosporium* species. * *p*-value < 0.01; ns: non-significant. Outliers are represented by an empty circle. TFC1: transcription protein on polymerase III promoters; UbcB: ubiquitin carrier protein; Sac7: rho guanosine triphosphatase activator; Fis1: mitochondrial membrane fission protein; SarA: guanosine triphosphate-binding protein; UBC6: protein catabolic process; Act: Actin; Tub: *β*-tubulin; GAPDH: D-glyceraldehyde-3-phosphate dehydrogenase.

**Figure 4 jof-09-00105-f004:**
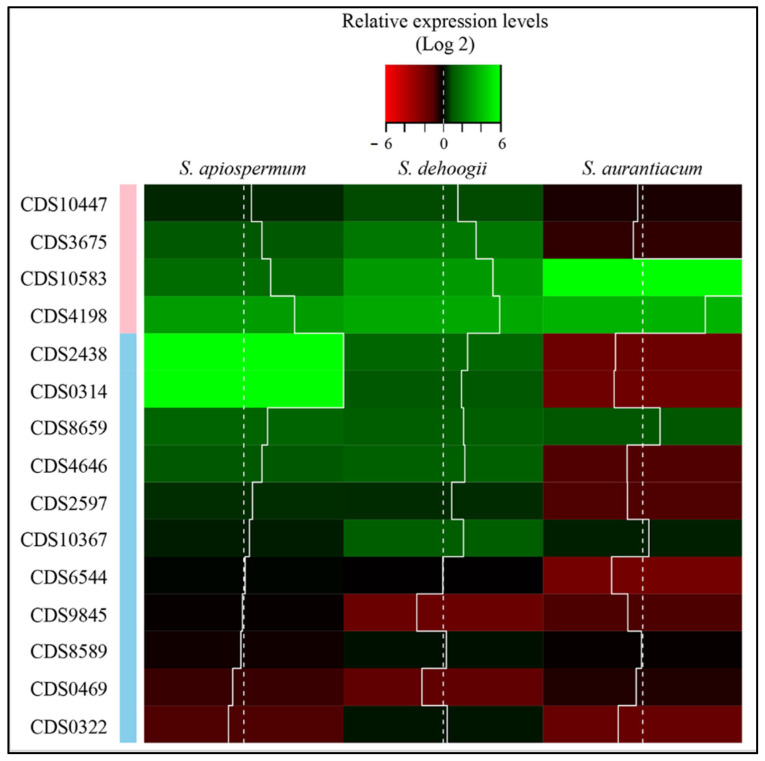
Relative expression level (Log2) of candidate genes in the presence of lignin. For each species, results correspond to the average of RT-qPCR analysis performed on three different isolates. Pink: peroxidases; blue: oxidases. CDS: coding sequence.

**Figure 5 jof-09-00105-f005:**
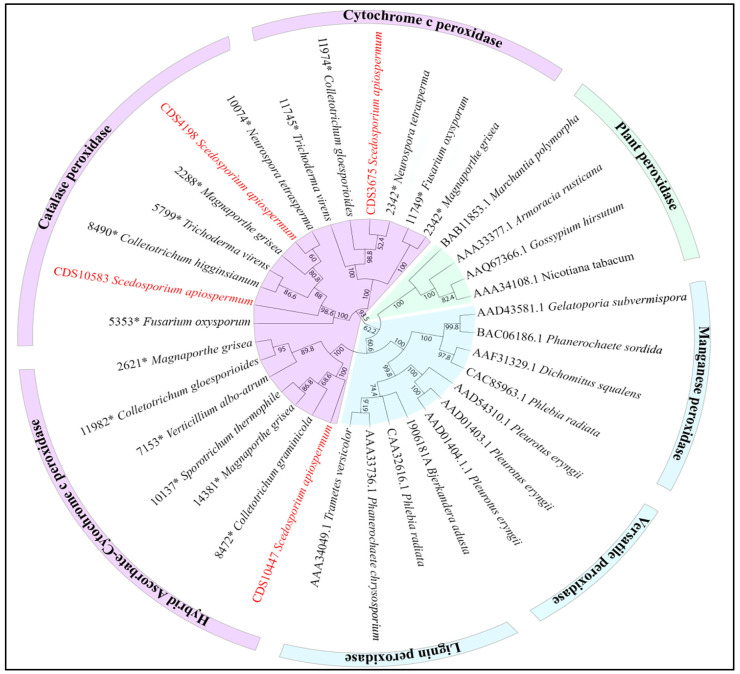
Classification of candidate enzymes among fungal peroxidases. An alignment of the protein sequences was created using the default settings for multiple alignments. Phylogenetic trees were constructed by the neighbor-joining method with cladogram presentation. Bootstrapping was carried out with 500 replications. Class I peroxidases are colored in pink, class II in blue and class III in green. Genes studied within *Scedosporium* species are marked in red. * Identification number from RedoxBase.

**Figure 6 jof-09-00105-f006:**
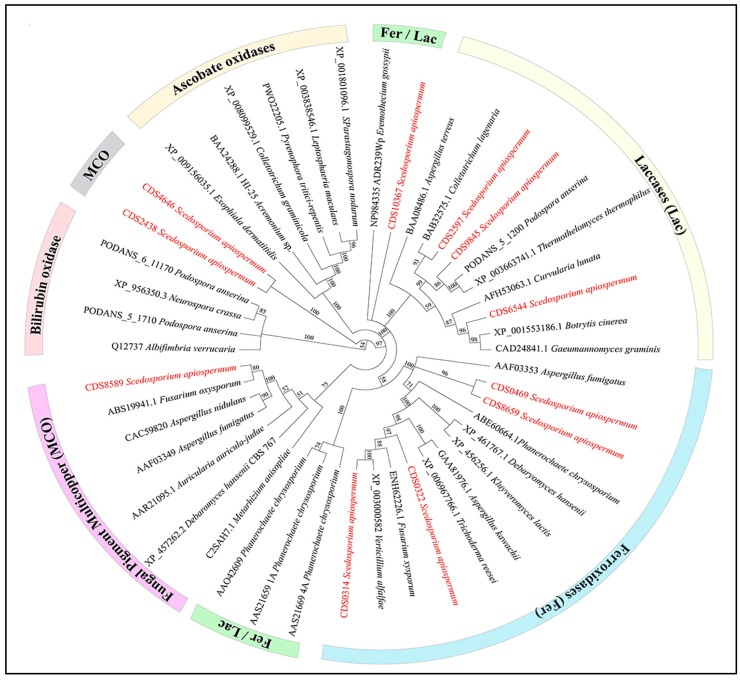
Classification of candidate enzymes among fungal MCO. An alignment of the protein sequences was created using the default settings for multiple alignments. Phylogenetic trees were constructed by the neighbor-joining method with cladogram presentation. Bootstrapping was carried out with 500 replications. Ferroxidases (Fer) are colored in blue, Laccases (Lac) in light yellow, Ferroxidases/Laccases (Fer/Lac) in green, Ascorbate oxidases in light orange, Multicopper oxidases (MCO) in grey, Bilirubin oxidase in dark orange and Fungal pigment multicopper in purple. Genes studied within *Scedosporium* species are marked in red.

## Data Availability

All the genome sequences used are publicly available and the WGS ID were provided in the manuscript.

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
