# Peer review of "Lignin-Modifying Enzymes in Scedosporium Species"

_jof, 2023, doi:10.3390/jof9010105_

Round 1
Reviewer 1 Report
In the article entitled “Lignin-modifying enzymes in Scedosporium species” the authors intent to demonstrate that Scedosporium species are able to utilize lignocellulose compounds as carbon source. The study is very important since those results can indirectly suggest the natural habitat of Scedosporium, besides add value information about the lignocellulose degradation by Ascomycetes. The study is very well conducted, but I have some points to discuss:
· Why the authors compared the growth studies in Scedo-Select III containing different carbon sources with YPDA? As the composition of YPDA is very different from Scedo-Select III the growth comparison is difficult.
· How the fungal inoculum of growth studies on agar was prepared? This information should be add to the Materials and Methods section.
· I think would be beneficial to the readers if the authors explain if there is differences in the degradation process of cellulose, xylan, and lignin.
· I think would also be beneficial to the readers if the authors add information about the lower growth on Scedo-Select III containing glucose as carbon source compared to xylan, for example.
· Why the authors evaluated the gene expression after only 4-h of growth in lignin presence? A higher time of incubation could lead to an increase in some gene expression?
· Why the authors made a pre-incubation of conidia cells in YEPD medium previously the incubation in Scedo-Select III containing lignin? And why the inoculum was made with germ tubes and not conidia?
· The abbreviation “MCO” is used in page 3, but its definition is only found at page 7, please correct.
· Why the authors did not discuss the results found herein with the previously analysis performed by the same research group in the article “Poirier W, Ravenel K, Bouchara JP, Giraud S. Lower Funneling Pathways in Scedosporium Species. Front Microbiol. 2021 Jul 2;12:630753. doi: 10.3389/fmicb.2021.630753. PMID: 34276578; PMCID: PMC8283699.”? For example, the use of lignin as the carbon source by S. apiospermum and S. aurantiacum was previously described in the aforementioned article, but is not mentioned in the introduction or discussion section.
Author Response
In the article entitled “Lignin-modifying enzymes in Scedosporium species” the authors intent to demonstrate that Scedosporium species are able to utilize lignocellulose compounds as carbon source. The study is very important since those results can indirectly suggest the natural habitat of Scedosporium, besides add value information about the lignocellulose degradation by Ascomycetes. The study is very well conducted, but I have some points to discuss:
We thank the reviewer for his/her positive comments which help us to improve our manuscript.
- Why the authors compared the growth studies in Scedo-Select III containing different carbon sources with YPDA? As the composition of YPDA is very different from Scedo-Select III the growth comparison is difficult.
We agree with the reviewer that it appears difficult to compare growth of a fungus on YPDA culture medium containing the easily assimilated carbon source that is glucose, to its growth on a minimal culture medium containing polymeric substrates such as cellulose, xylan or lignin as the sole carbon and energy source. Nevertheless, YPDA is a gold standard in medical mycology. It was therefore used here as a reference culture medium, to strengthen our data showing the ability of Scedosporium species to use the complex cellulose, xylan or lignin substrates as the carbon source since mean diameters of the colonies on these substrates always were greater than 50% of those obtained on YPDA and sometimes quite equivalent. The corresponding sentence were reformulated in the Materials and Methods section (l. 87-88).
“Results were compared with those obtained on the rich and commonly used medium YPDA.”
- How the fungal inoculum of growth studies on agar was prepared? This information should be add to the Materials and Methods section.
As suggested by the reviewer, a sentence was added in the Material and Methods section to explain the preparation of the fungal inoculum (l. 84-85).
« Mycelium from a 14-day-old YPDA culture was collected using a sterile needle and inoculated by central pricking. »
- I think would be beneficial to the readers if the authors explain if there is differences in the degradation process of cellulose, xylan, and lignin.
According to this comment, some sentences were added in the Introduction section to explain the differences in the degradation process of cellulose and hemi-cellulose (xylans) versus lignin (l. 25-37).
“Fungi are key players in the degradation of lignocellulosic biomass. Besides cellulose and hemicelluloses (mainly xylans), lignin is a macromolecule providing strength as well as rigidity to the plant cell wall and protecting its two polysaccharidic partners against microbial attack by hydrolytic enzymes. Whereas degradation of cellulose and hemicellulose involves glycosidases (mainly cellobiohydrolases, cellulose cellobiosidases and glycosydases, or xylanases and xylohydrolases, respectively), degradation of lignin is more complex as it is a polyphenolic macromolecule resulting from polymerization of three types of monolignols (paracoumarylic, coniferylic and sinapylic acids). As a consequence, whereas many microorganisms are able to use cellulose and hemicellulose as the carbon source, only a few groups of fungi are able to degrade lignin, among which the wood-decaying basidiomycetes, also termed “white-rot fungi”, are the most efficient. They mineralize the complex lignin polymer by the synergistic action of several extracellular enzymes, such as peroxidases and laccases [1].”
- I think would also be beneficial to the readers if the authors add information about the lower growth on Scedo-Select III containing glucose as carbon source compared to xylan, for example.
This part was reformulated in order to take into account this comment as well as the general comment of the second reviewer considering the statistical analysis (l. 182-188).
“The fungus was not able to develop in the absence of any carbon source (data not shown), and it grew more slowly on a lignin-containing synthetic medium compared to the other culture media (Figure 1a and supplemental figure). In addition, for most of the strains, the mean diameter of the colonies after ten days of incubation was significantly higher in the presence of cellulose or xylan compared to the results obtained in the presence of lignin or, strikingly, of glucose (Figure 1b). »
Strikingly, compared to glucose, the presence of complex substrates seems to stimulate growth of Scedosporium species. We may hypothesize that to obtain glucose from complex substrates like cellulose and xylan, the fungus is forced to produce degradation enzymes and to overexpress genes encoding enzymes involved in the oxidative degradation of glucose, thus leading to a higher production of ATP. However, this is only a hypothesis that could be investigated, and in the absence of experimental evidence, it seems difficult to mention it in the Discussion section.
- Why the authors evaluated the gene expression after only 4-h of growth in lignin presence? A higher time of incubation could lead to an increase in some gene expression?
As the medium contained a unique carbon source, the fungus should adapt as soon as possible. A preliminary study was performed on few strains, aiming to identify the most appropriate duration of incubation (between 4 and 24 h) to reveal over-expression of the target genes. The best results were obtained with a 4 h-incubation time, which was selected for the experiments reported in our manuscript.
- Why the authors made a pre-incubation of conidia cells in YEPD medium previously the incubation in Scedo-Select III containing lignin? And why the inoculum was made with germ tubes and not conidia?
We understand that this pre-incubation may appear surprising and requires some explanations. Actually, the kinetic of germination is different according to the carbon source and genes involved in degradation of cellulose, xylan or lignin should be expressed following the germination step, during the hyphal growth. To evaluate the impact of the carbon source on the expression level of the target genes, a short duration of incubation (and of course the same for all carbon sources) is needed to avoid degradation of mRMAs. Therefore, direct inoculation of cellulose, xylan or lignin containing Scedo-Select III culture media with conidia was not possible. In order to normalize this initial step of germination, Scedosporium spores were first incubated in YEPD, and the culture medium was then changed to evaluate the impact of the carbon source.
Some sentences therefore were added in the Materials and Methods section in order to explain it (l. 94-98)
“To evaluate the impact of the carbon source on the expression level of the target genes, a short duration of incubation (the same for all carbon sources) was needed to avoid degradation of mRMAs. As the germination kinetic vary according to the carbon source, a normalization of the germination step therefore was performed by preincubation of the conidia in YPD medium for 24 h in order to obtain germ-tubes. Briefly, 2.107 conidia were inoculated in 50 mL of YEPD medium … »
- The abbreviation “MCO” is used in page 3, but its definition is only found at page 7, please correct.
We apologize for this mistake. The change was done (l. 150).
- Why the authors did not discuss the results found herein with the previously analysis performed by the same research group in the article “Poirier W, Ravenel K, Bouchara JP, Giraud S. Lower Funneling Pathways in Scedosporium Species. Front Microbiol. 2021 Jul 2;12:630753. doi: 10.3389/fmicb.2021.630753. PMID: 34276578; PMCID: PMC8283699.”? For example, the use of lignin as the carbon source by S. apiospermum and S. aurantiacum was previously described in the aforementioned article, but is not mentioned in the introduction or discussion section.
A sentence was added in the discussion section in this way (l. 265-268).
“In a previous work, we demonstrate that Scedosporium species display all the genetic equipment needed for the intracellular degradation of phenolics compounds. These lower funneling pathways degrade aromatic compounds, like those resulting from the extracellular degradation of lignin.”
Reviewer 2 Report
Thanks for the opportunity to review the manuscript titled “Lignin-modifying enzymes in Scedosporium species” by Wilfried Poirier, Jean-Philippe Bouchara and Sandrine Giraud. The manuscript addresses an issue that is very relevant to a better understanding of the Scedosporium complex, as much needs to be known about this fungal species. However, this manuscript is not ready for publication yet. There are some issues that the authors need to address before the manuscript can be considered for publication. The following are my comments describing these issues:
- In the Abstract (lines 6-8), the authors should revise the reference of the species as belonging to the genus Scedosporium, because it is not accurate anymore. It is actually considered a complex, even more if we relate these species to the cystic fibrosis infection, as described for many authors, such as Noni and collaborators (Mycoses. 2017;60:594–599): “Scedosporium apiospermum complex comprises five closely related species, S. apiospermum sensus stricto, Scedosporium boydii, Scedosporium aurantiacum, Scedosporium dehoogii and Scedosporium minutisporum. They are the second most frequent filamentous fungi that can be found in cystic fibrosis (CF) patients and prevalence is reported as 0.7%-9%.”
- Figure 1: (a) The symbols are very small, what makes difficult to read the growth curves, even more because 3 of them are in black. (b) The shadows of gray for different conditions also make more difficult to understand the bars. As the graphs are already in color, I suggest to choose different color to make easier the understanding of the data.
Still about the figure 1 – there is no statistic showing if there is a real difference between the conditions. It says there was a “slowly growth on a lignin- or glucose-containing synthetic medium, but similar Growth on xylan-containing medium when compared to YPDA” (lines 175-178). Statistics should give us this information in a better way. Please provide it.
Figure 1 (a): Is it possible to provide the kinetic data for the other 2 species, even as supplementary material?
- Figure 2 (a) and (b): same problem. Without the statistics and also with the huge standard deviation in the whole experiment, it is not possible to affirm something like “both enzymatic activities were higher when cultures performed in the presence of lignin (Figure 2). Except for S. dehoogii UA 110354504 isolate, the peroxidase activity was at least two-fold higher when the fungus was cultivated in a lignin-containing medium compared to control conditions. Similarly, a 2-fold increase was seen for laccase activity for 2 out of the 3 isolates per species in cultures grown in the presence of lignin” (lines 192-197). Please provide statistic numbers for this graphs/affirmations.
- Figure 3: “For each gene, an ANOVA test was performed to analyze the variations between Scedosporium species” (lines 213-214). Which type of ANOVA? Which post-test using ANOVA? A better description should be provide to describe the used statistic test.
In addition, figure 3 was very difficult to understand. The * was used to compare…what is the comparison from? It is significant in all the species? A better explanation of the data should be provided.
- Figure 5: I couldn’t understand very well whys this figure is in the Discussion and not in the Results section. Besides, it was very difficult to read this figure, only with a high zoom on the pdf file.
General Comment:
The manuscript needs to be edited for statistic methodology. In my opinion, it is a big and very problematic issue that makes the article not publishable yet. I thank you again for the opportunity to review this manuscript. My overall recommendation is going to be minor revision, although I really think the missing statistic should be considered a big issue. I highly recommend the authors to work on it and address it in the correct way for the paper final acceptance.
Author Response
Thanks for the opportunity to review the manuscript titled “Lignin-modifying enzymes in Scedosporium species” by Wilfried Poirier, Jean-Philippe Bouchara and Sandrine Giraud. The manuscript addresses an issue that is very relevant to a better understanding of the Scedosporium complex, as much needs to be known about this fungal species. However, this manuscript is not ready for publication yet. There are some issues that the authors need to address before the manuscript can be considered for publication. The following are my comments describing these issues:
- In the Abstract (lines 6-8), the authors should revise the reference of the species as belonging to the genus Scedosporium, because it is not accurate anymore. It is actually considered a complex, even more if we relate these species to the cystic fibrosis infection, as described for many authors, such as Noni and collaborators (Mycoses. 2017;60:594–599): “Scedosporium apiospermum complex comprises five closely related species, S. apiospermum sensus stricto, Scedosporium boydii, Scedosporium aurantiacum, Scedosporium dehoogii and Scedosporium minutisporum. They are the second most frequent filamentous fungi that can be found in cystic fibrosis (CF) patients and prevalence is reported as 0.7%-9%.”
We fully agree with the statements reported in this paper from Noni and collaborators published in 2017. However, taxonomy within the Scedosporium genus changed a lot in recent years. For instance, in 2018, the ECMM/ISHAM working group on Scedosporium and Lomentospora infections published a review on recent advances in this field, including taxonomy (Ramirez-Garcia A., Pellon A., et al. Scedosporium and Lomentospora: an updated overview of underrated opportunists, Medical Mycology, Medical Mycology, 2018, 56 (suppl.1) : 102-125). In this community paper, ten species were recognized in the Scedosporium genus and the term of complex was restricted to S. apiospermum, S. boydii, S. ellipsoideum, S. fusoideum and S. angustum. As S. aurantiacum and S. dehoogii (which no longer belong to the S. apiospermum complex, we could not use the term of complex, but only of Scedosporium species. In addition, whereas S. dehoogii is common in the environment, it has never been reported from patients with cystic fibrosis.
- Figure 1: (a) The symbols are very small, what makes difficult to read the growth curves, even more because 3 of them are in black. (b) The shadows of gray for different conditions also make more difficult to understand the bars. As the graphs are already in color, I suggest to choose different color to make easier the understanding of the data.
As growth curves are very close to each other, increasing the size of the symbols was not possible in Figure 1a. Nevertheless, to take into account the reviewer’s comment, different colors were used for each carbon source to increase the readability of Figure 1a, and the shadows of grey were changed in Figure 1b, for colors as suggested, using the same colors in Figures 1a and 1b for a particular substrate.
- Still about the figure 1 – there is no statistic showing if there is a real difference between the conditions. It says there was a “slowly growth on a lignin- or glucose-containing synthetic medium, but similar Growth on xylan-containing medium when compared to YPDA” (lines 175-178). Statistics should give us this information in a better way. Please provide it.
Significance of the results was confirmed by statistical analysis and this information was added on the figure (using asterisks), as well as in the Materials and Methods section (l. 88-89).
“For statistical analyses, an ANOVA-Tukey test was performed to compare growth at 10 days within each culture conditions, * p-value < 0.05.”
- Figure 1 (a): Is it possible to provide the kinetic data for the other 2 species, even as supplementary material?
A supplemental figure was added, containing the growth kinetic of one selected strain for the two other species studied.
- Figure 2 (a) and (b): same problem. Without the statistics and also with the huge standard deviation in the whole experiment, it is not possible to affirm something like “both enzymatic activities were higher when cultures performed in the presence of lignin (Figure 2). Except for S. dehoogii UA 110354504 isolate, the peroxidase activity was at least two-fold higher when the fungus was cultivated in a lignin-containing medium compared to control conditions. Similarly, a 2-fold increase was seen for laccase activity for 2 out of the 3 isolates per species in cultures grown in the presence of lignin” (lines 192-197). Please provide statistic numbers for this graphs/affirmations.
Statistical analysis was performed and this part of the result section was modified (l. 202-206).
“There was an important inter-experimental variability (possibly explained by the fact the analysis was conducted during the early phase of hyphal growth – after 24 h of incubation) leading to important standard deviations. Significant differences were observed for three of the strains tested regarding the peroxidase activity and three other strains for laccase activities. Nevertheless, for all the species both enzymatic activities tend to increase in the presence of lignin as carbon source (Figure 2).”
A sentence was also added in the Material and Methods.
“According to the data distribution, statistical analysis was performed using a Student-T or Mann-Whitney test.” (l. 113-114)
- Figure 3: “For each gene, an ANOVA test was performed to analyze the variations between Scedosporium species” (lines 213-214). Which type of ANOVA? Which post-test using ANOVA? A better description should be provide to describe the used statistic test.
A Tukey-ANOVA test was chosen to compare species between them for each of the target genes. This information was added in the legend of the figure and in the Materials and Methods section. (l. 176)
- In addition, figure 3 was very difficult to understand. The * was used to compare…what is the comparison from? It is significant in all the species? A better explanation of the data should be provided.
Figure 3 was modified in order to clarify the significant differences observed between the three species for each of the genes studied.
- Figure 5: I couldn’t understand very well whys this figure is in the Discussion and not in the Results section. Besides, it was very difficult to read this figure, only with a high zoom on the pdf file.
In order to increase the readability, Figure 5 was divided into two figures (corresponding to fungal peroxidases and to MCO, respectively), which permitted to enlarge them.
However, as these figures 5 and 6 illustrate the idea developed in the Discussion section that Scedosporium spp can be classified among the soft-rot fungi, it seems to us more appropriate to maintain them in the Discussion section, rather than to move to the Results section which only presents the identified peroxidases and oxidases activities and gene expressions.
General Comment:
The manuscript needs to be edited for statistic methodology. In my opinion, it is a big and very problematic issue that makes the article not publishable yet. I thank you again for the opportunity to review this manuscript. My overall recommendation is going to be minor revision, although I really think the missing statistic should be considered a big issue. I highly recommend the authors to work on it and address it in the correct way for the paper final acceptance.
We fully agree with the reviewer regarding the positive impact of these statistical analyses. We take into account this general comment and statistical analyses were added to the figures.